# Simultaneous Monitoring of pH and Chloride (Cl^−^) in Brain Slices of Transgenic Mice

**DOI:** 10.3390/ijms222413601

**Published:** 2021-12-18

**Authors:** Daria Ponomareva, Elena Petukhova, Piotr Bregestovski

**Affiliations:** 1Institut de Neurosciences des Systèmes, Aix-Marseille University, INSERM, INS, 13005 Marseille, France; Ponomareva_DN@mail.ru; 2Institute of Neurosciences, Kazan State Medical University, 420111 Kazan, Russia; petukhovaeo@mail.ru; 3Department of Normal Physiology, Kazan State Medical University, 420111 Kazan, Russia

**Keywords:** genetically encoded biosensors, optopharmacology, transgenic mice, intracellular pH, intracellular chloride, brain slices, pH and Cl^−^ transporters

## Abstract

Optosensorics is the direction of research possessing the possibility of non-invasive monitoring of the concentration of intracellular ions or activity of intracellular components using specific biosensors. In recent years, genetically encoded proteins have been used as effective optosensory means. These probes possess fluorophore groups capable of changing fluorescence when interacting with certain ions or molecules. For monitoring of intracellular concentrations of chloride ([Cl^−^]_i_) and hydrogen ([H^+^] _i_) the construct, called ClopHensor, which consists of a H^+^- and Cl^−^-sensitive variant of the enhanced green fluorescent protein (E^2^GFP) fused with a monomeric red fluorescent protein (mDsRed) has been proposed. We recently developed a line of transgenic mice expressing ClopHensor in neurons and obtained the map of its expression in different areas of the brain. The purpose of this study was to examine the effectiveness of transgenic mice expressing ClopHensor for estimation of [H^+^]_i_ and [Cl^−^]_i_ concentrations in neurons of brain slices. We performed simultaneous monitoring of [H^+^]_i_ and [Cl^−^]_i_ under different experimental conditions including changing of external concentrations of ions (Ca^2+^, Cl^−^, K^+^, Na^+^) and synaptic stimulation of Shaffer’s collaterals of hippocampal slices. The results obtained illuminate different pathways of regulation of Cl^−^ and pH equilibrium in neurons and demonstrate that transgenic mice expressing ClopHensor represent a reliable tool for non-invasive simultaneous monitoring of intracellular Cl^−^ and pH.

## 1. Introduction

Neuronal activity is accompanied by dynamical changes in the intra- and extracellular concentration of ions. A number of studies demonstrated that activation or inhibition of neurons can cause a rapid shift of hydrogen (H^+^) and chloride (Cl^−^) [1,2,3]. The physiological intracellular pH range in different eukaryotic cells is 6.5–8.0 [4], which corresponds to a very low free H^+^ concentration, from 10 nM to 300 nM. In the mammalian brain the intracellular pH is 7.0–7.4 [5,6] reflecting the importance of maintaining the intracellular H^+^ concentration ([H^+^]_i_) in a narrow range.

Intracellular Cl^−^ concentration ([Cl^−^]_i_) in different cell types varies from 3 mM to 60 mM, being around 5–10 mM in the majority of mammalian neurons [7]. Deviations from this physiological range can alter the excitability of cells, modulate the function of a variety of proteins including ion channels [8,9,10]. Abnormal changes in concentrations of these ions are associated with the development of pathological processes and some disorders including neurodegeneration, epilepsy and brain ischemia [11,12,13].

To maintain H^+^ and Cl^−^ in physiological ranges, various transporters, cotransporters, and other ion regulating proteins are expressed in cells of biological organisms. The regulation of intracellular [H^+^]_i_ is primarily driven by Na^+^/H^+^ exchange, Na^+^-driven Cl^−^/HCO_3_^−^ exchange, Na^+^/HCO_3_^−^ cotransport, and Cl^−^/HCO_3_^−^ exchange [1,14,15,16].

The level of [Cl^−^]_i_ is maintained predominantly by a K^+^/Cl^−^ cotransporter (KCC2), which pumps out Cl^−^ from the cells, a Na^+^/K^+^/Cl^−^ cotransporter NKCC1, which loads Cl^−^ into the cell, and a Cl^−^/HCO_3_^−^ exchanger [3]. Activation of the Cl^−^/HCO_3_^−^ exchanger leads to local simultaneous pH and Cl^−^ changes. These interrelated relationships highlight the necessity of simultaneous monitoring of changes in Cl^−^ and H^+^ concentrations.

To measure pH and Cl^−^ in cells of biological organisms, several methods have been proposed. The most used are ion-selective microelectrodes [17,18,19], fluorescent dyes [20,21,22] and genetically encoded probes [7,23,24,25,26]. Especially promising are genetically encoded fluorescent sensors, which allow non-invasive monitoring of intracellular ion concentrations in specific cell types.

For simultaneous registration of [H^+^]_i_ and [Cl^−^]_i_, the construct ClopHensor has been developed, which consists of a H^+^- and Cl^−^ -sensitive variant of the enhanced green fluorescent protein (E^2^GFP) fused with a monomeric red fluorescent protein (mDsRed) [27]. The sensor was studied at the heterologous expression in different cells. It has been shown that the construct possesses a pKa = 6.8 for H^+^ and K_d_ for Cl^−^ in the range 40–50 mM at physiological pH ~ 7.3 [24,27,28]. Improved variants of ClopHensor have been developed and tested in cultured brain slices using biolistical transfection [29]. Another biosensor optimized for the simultaneous [Cl^−^]_i_ and pH_i_ imaging, called LSSmClopHensor, has been developed [30]. It was expressed in the cortex of rats using in utero electroporation and used for analysis in brain slices and in vivo [30,31].

Recently, we presented a line of transgenic mice expressing ClopHensor in neurons due to the neuronal-specific promoter Thy1 [32]. The pattern of ClopHensor expression across the brain has been revealed using the CLARITY method in combination with confocal and light-sheet microscopy. It showed a robust expression of ClopHensor in the hippocampal formation, thalamus, ponds, medulla, cerebellum and other areas [32].

The present study is devoted to the analysis of H^+^- and Cl^−^-transients at synaptic stimulation of neurons in hippocampal slices of Thy1: ClopHensor mice. We analyzed changes in both ion concentrations at different approaches including inhibition of GABA-ergic synaptic transmission, extracellular Ca^2+^-free and low Cl^−^ or Na^+^ conditions. Our observations demonstrate the efficiency of transgenic mice expressing ChlopHensor for reliable non-invasive monitoring of intracellular Cl^−^ and pH in normal and pathological conditions.

## 2. Results

### 2.1. Simultaneous Monitoring of pH and Cl^−^ Using ClopHensor

Experiments were performed on brain slices of transgenic mice expressing ClopHensor using the mouse neuronal promoter Thy1. ClopHensor consists of a modified enhanced green fluorescent protein E^2^GFP connected with a monomeric DsRed (mDsRed) via a 20-amino-acid flexible linker (Figure 1A). E^2^GFP carries a specific anion-binding site engineered by the single substitution T203Y and elevation of Cl^−^ causes static quenching of E2GFP’s fluorescence [33]. Like all green fluorescent proteins, E^2^GFP is also sensitive to pH: its emission decreases when it is acidified. However, E^2^GFP’s emission intensity does not depend on pH when excited at its isosbestic point of 458 nm, allowing one to perform ratiometric analysis. In addition, at wavelengths above 540 nm, signals of both fluorescent proteins, E^2^GFP and mDsRed, are pH and Cl^−^-independent (Figure 1B). This allows the simultaneous ratiometric analysis of changes in [H^+^]_i_ and [Cl^−^]_i_ at monitoring upon excitation at three wavelengths: 488 nm: pH- and Cl^−^-dependent E^2^GFP signal, 458 nm: pH-independent E^2^GFP signal, and above 543 nm: Cl^−^- and pH-independent mDsRed signal [27].

In the present study, we recorded fluorescence at slightly different excitation wavelengths (455 nm, 505 nm and 590 nm) and estimated changes in concentrations of ions using the following ratios:R_pH_ = ΔF_505nm_ / ΔF_455nm_(1)
R_Cl_ = ΔF_590nm_ / ΔF_455nm_(2)

Examples of fluorescence at different excitation wavelengths are shown in Figure 1C. Details of calibration are presented in Section 4.

### 2.2. Effect of Bicuculline on Synaptically Induced Cl^−^ and pH-Transients in Hippocampal Neurons

To examine the properties of intracellular Cl^−^ and H^+^ transients during synaptic activation, we performed experiments on hippocampal slices of transgenic mice expressing ClopHensor. The first task was to establish how the changes in intracellular concentrations of Cl^−^ and H^+^, evoked by electrical stimulation, are amenable to pharmacological modulation of synaptic transmission. Fluorescent emission signals were recorded from the hippocampal CA1 area upon sustained tetanic stimulation of Schaffer’s collaterals (100 Hz for 20 s). Stimulus intensities ranged from 20 μA to 320 μA. The experiments were carried out on adult transgenic mice aged 2–8 months.

High-frequency stimulation of Schaffer’s collaterals caused strong changes of fluorescence signals excited at 455 nm and 505 nm without effect on excitation at 590 nm (not shown). Ratiometric analysis revealed that these changes correspond to an elevation of both [Cl^−^]_i_ and [H^+^]_i_. In this set of experiments, in hippocampal CA1 neurons, the base level of [Cl^−^]_i_ was 8.0 ± 1.1 mM (*n* = 5) and the mean intracellular pH was 7.30 ± 0.02 (*n* = 11). High frequency synaptic stimulation increased the [Cl^−^]_i_ to 0.8 ± 0.3 mM (*n* = 5) and acidification of neurons to 0.024 ± 0.006 units pH (*n* = 11).

In order to test the assumption that during synaptic stimulation, the accumulation of Cl^−^ in the hippocampal neurons is, at least partially, due to Cl^−^ influx via activated GABAA receptors, we applied bicuculline, an antagonist of these receptors [34]. The traces shown in Figure 2A illustrate that upon the synaptic stimulation, bicuculline strongly decreased the amplitude of synaptically induced changes of [Cl^−^]_i_. On average, the mean amplitude of Cl^−^ influx decreased to 41.2 ± 12.2% (Figure 2B, *n* = 7, *p* < 0.01). In contrast, upon application of bicuculline the amplitude of synaptically induced pH transients increased by 90.7 ± 17.4% (Figure 2C, *n* = 8, *p* < 0.01). Picrotoxin, a blocker of Cl^−^-selective GABA receptor channels, caused similar changes in the amplitudes of Cl^−^ and pH transients (data not shown).

The augmentation of pH transients can be, at least partially, explained by the fact that bicuculline blocks inhibitory transmission, which leads to an increase in the general excitability [35]. To check the involvement of this mechanism in the potentiation of synaptically induced H^+^ responses upon inhibition of GABA receptors, we analyzed the effect of bicuculline on the amplitude of evoked local field potentials (eLFP). After stimulation of Schaffer’s collaterals, a population of CA1 neurons are activated simultaneously and fire an action potential in synchrony, giving rise to a single eLFP. As illustrated in Figure 2D, the addition of bicuculline (40 µM) led to an elevation of eLFPs. The mean potentiation was 24 ± 3.8% (Figure 2E). Consequently, in the presence of bicuculline, the effectiveness of the stimulation increased, causing stronger depolarization and augmentation of pH transients. A competitive AMPA/kainate receptor antagonist (CNQX, 40 µm) completely abolished stimulation-induced changes of ratiometric emission signals, confirming that they have emerged from synaptic activation (not shown).

Altogether, these data demonstrate that: (i) bicuculline modulates in opposite ways the synaptically evoked Cl^−^- and pH-transients in neurons expressing ClopHensor; (ii) bicuculline only partially inhibits synaptically induced elevation of [Cl^−^]_i_; (iii) inhibition of GABA receptor by bicuculline causes potentiation of eLFPs.

### 2.3. Effect of Bicuculline on Intracellular pH Changes after Tetanic Stimulation, Assessed Using BCECF-AM

To ensure the specificity of the pH changes when using ClopHsensor, we performed a series of experiments with the pH-sensitive dye BCECF-AM [36]. This compound is well tolerated by cells and offers ratiometric estimation of pH values.

We determined the pH changes by calculating the ratio of two emission signals obtained after illumination by light at 455 nm and 505 nm. The intracellular pH changes after tetanic stimulation and its modulation by bicuculline were analyzed. Experiments were conducted on wild type juvenile mice (P10–P12). Registration conditions were similar to those performed on slices from transgenic ClopHensor mice. Examples of CA1 hippocampal neurons loaded with BCECF and excited at different wavelengths are presented in Figure 3A. Evoked pH-specific ratiometric emission signals were obtained by giving the same pattern of electrical stimulation (100 Hz for 20s); the stimulus intensities ranged from 200 μA to 250 μA.

Similar to the observations with ClopHensor, on neurons loaded with BCECF-AM, the addition of bicuculline resulted in an increase of the pH transients evoked by tetanic stimulation (Figure 3B). Summary data from 5 slices showed that upon addition of bicuculline, the mean amplitude of the pH signals increased to 208.6 ± 48.8% (Figure 3C, *n* = 5, *p* < 0.05).

These results demonstrate that: (i) sustained high-frequency stimulation induces acidification of cells and bicuculline potentiates the amplitude of pH transients; (ii) pH-specific emission signals are similar at recording either with BCECF-AM or using ClopHensor.

### 2.4. Analysis of the Decay Kinetics of Cl^−^ and pH Transients

We then performed a comparative analysis of the kinetics of synaptically induced Cl^−^ and pH transients. It was evaluated by determining the decay time constants, i.e., the time during which the peak amplitude of Cl^−^ and H^+^ components decreases by e-times (τ_decay_).

The τ_decay_ of pH-fluorescent signals recorded from hippocampal slices of mice expressing ClopHensor was 157 ± 20.3 s, *n* = 16 (Figure 4A,B). This was close to values obtained on slices from wild type mice loaded with BCECF (τ_decay_=188.6 ± 16.5 s, *n* = 14; Figure 4A,B). The concentration of intracellular Cl^−^ ions was restored considerably faster than the pH after high-frequency stimulation of synaptic inputs (Figure 4A, red trace). The mean τ decay of Cl^−^ transients was 35 ± 4.2 s (*n* = 9, Figure 4B), i.e., about 4.5 times shorter than that of pH signals.

These results suggest that the mechanisms involved in the withdrawing of ions from neurons operate much faster for Cl^−^ than for H^+^ and different mechanisms are involved in the maintenance of normal physiological transmembrane concentrations of these ions.

Next, we analyzed how changes of physiologically important ions, involved in regulation of Cl^−^ and H^+^ transporters, modulate the amplitude of Cl^−^ and pH transients induced by depolarization.

### 2.5. Changes of [H^+^]_i_- and [Cl^−^]_i_-Induced by High [K^+^]_o_-Depolarization

Several studies used [K^+^]_o_-induced depolarization as a tool for stimulation of pH changes in different cell types [37,38,39,40]. We used this approach to analyze the properties of pH_i_ and [Cl^−^]_i_ transients in neurons of hippocampal slices under different experimental conditions.

In all experiments, the high [K^+^]_o_ solution contained 20 mM, i.e., 17.5 mM of potassium gluconate was added to normal ASCF containing 2.5 mM K^+^. Depolarization induced by application of high [K^+^]_o_ ASCF caused a remarkable elevation of [Cl^−^]_i_ in all slices, with an average of 11.4 ± 1.1 mM (*n* = 11). Concerning pH, two types of responses were observed. In the majority of slices, [K^+^]_o_-induced depolarization caused acidification of neurons by, on average, 0.15 ± 0.02 pH units (*n* = 23). In 8 cases, biphasic pH_i_-shifts composed of an early short-lasting alkalinisation that turned into a longer-lasting acidification were observed. We have not analyzed yet the reasons for this difference in the pH responses.

### 2.6. [K^+^]_o_-Induced Changes of pH and Cl^−^ in a Ca^2+^-Free ACSF

Observations of cells in culture conditions demonstrated that elevation of external K^+^ causes strong depolarization of neurons, accompanied, in addition to acidification, by a rise in intracellular Ca^2+^ [38,39]. In these studies, acidic [K^+^]_o_-induced pH responses were either completely prevented by the use of a Ca^2+^-free medium [38], or even inversed and became alkalizing [39], suggesting a key role of Ca^2+^ ions in depolarization-induced pH changes.

We performed an analysis of the removal of external Ca^2+^ on depolarization-induced transients of [Cl^−^ ]_i_ and pH_i_ in brain slices of ClopHensor mice. Under control conditions, when high K^+^ was applied to slices for 3–6 min, an elevation in [Cl^−^]_i_ and decrease in pH were observed (Figure 5A). In this set of experiments, [K^+^]_o_-induced intracellular acidification occurred, on average by 0.12 ± 0.11 pH (*n* = 4, Figure 5C), and an increase in [Cl^−^]_i_ occurred by 10.12 ± 1.21 mM (*n* = 4, Figure 5B). After washing, [Cl^−^]_i_ recovered toward the baseline, while the base level of pH_i_ sometimes followed to more alkaline values (Figure 5A).

Applying a Ca^2+^-free ACSF containing 1 mM EGTA did not change the base level of [Cl^−^]_i_, while a decrease in pH, i.e., acidification of the cell cytoplasm was observed (Figure 5A). On average, the base level of pH_i_ decreased by 0.09 ± 0.02 рН (Figure 5D, *n* = 8, *p* < 0.01). [K^+^]_o_-induced depolarization caused an elevation of [Cl^−^]_i_ by 12.75 ± 0.83 mM (*n* = 4, Figure 5B). Surprisingly, unlike reports on preparations in culture [38,39], in our conditions, the pH responses to high K^+^ were acidifying, only with reduced amplitudes compared to the control. On average, the [K^+^]_o_-induced pH decrease was by 0.10 ± 0.08 pH (*n* = 4, Figure 5C).

### 2.7. The Effect of Extracellular Cl^−^ on Synaptically-Induced Changes of [Cl^−^]_i_ and pH_i_

Our next task was to determine the contribution of Cl^−^ ions to the [Cl^−^]_i_ and pH_i_ transients. For this, we analyzed the effects of decreasing Cl^−^ concentration in ACSF from 134.7 mM to 7.2 mM (“low Cl^−^” conditions). The changes in base levels of pH_i_ and [Cl^−^]_i_, as well as the amplitudes of Cl^−^ and H^+^ transients evoked by a high-frequency stimulation, were determined. On average, the mean amplitude of Cl^−^ influx on high-frequency stimulation was 0.8 ± 0.3 mM (Figure 6A,B). At the same time, stimulation caused acidification by 0.012 ± 0.005 pH (Figure 6A,C). After adding the “low-Cl^−^” ACSF, a diminishing base level of [Cl^−^]_i_, synchronously with alkalinisation of neurons, was observed. New levels of [Cl^−^]_i_ and pH_i_ were completely stabilized in about 15 min (Figure 6A). On average, [Cl^−^]_i_ diminished by 7.1 ± 1.1 mM (*p* < 0.05, *n* = 5) and рН_i_ increased by 0.19 ± 0.01 рН units (*p* < 0.05, *n* = 5; Figure 6D,E).

During the first minutes after establishing new steady-state levels in “low-Cl^−^” external solution, high-frequency stimulation of Schaffer’s collaterals caused the acidification directed pH responses, in which the amplitude was much higher than in control ACSF (Figure 6A). On average, ΔpH_i_ increased by 5 times in comparison with that recorded in the normal ACSF and became 0.063 ± 0.013 units pH (*p* < 0.05, *n* = 5, Figure 6C). The Cl^−^ transients on stimulation were nearly completely suppressed (Figure 6A,B). With an increase in the duration of incubation of slices in the “low-Cl^−^” ACSF, the amplitude of the evoked pH responses decreased until they completely disappeared (data not shown). Our preliminary observations suggest that the reason for this elimination of pH responses is a continuous strong decrease in the amplitude of evoked local field potentials upon transition to “low-Cl^−^” ACSF conditions.

### 2.8. Changes of [Cl^−^]_i_ and pH_i_ during [K^+^]_o_-Induced Depolarization in the Low-Cl^−^ ACSF

We then tested how a decrease in extracellular Cl^−^ affects the amplitude of [Cl^−^]_i_ and pH_i_ changes caused by the [K^+^]_o_-induced depolarization. Similarly to previously described observations (Section 2.5), the addition of 17.5 mM K^+^ in normal ACSF induced reversible transients of acidification and synchronous elevation of [Cl^−^]_i_ in neurons (Figure 7A). In this set of experiments, the mean K^+^-induced decrease in pH_i_ was 0.15 ± 0.02 pH and the elevation of [Cl^−^]_i_ was 4.36 ± 0.77 mM (*n* = 5).

As previously described, upon perfusion of slices with “low Cl^−^” ACSF, the base level of pH increased and [Cl^−^]_i_ synchronously decreased (Figure 7A). In “low Cl^−^” ACSF conditions, depolarization caused by the addition of 17.5 mM K^+^, resulted in a significant increase in the amplitude of pH acidification responses, while [Cl^−^]_i_ responses were nearly completely suppressed or even oppositely directed, i.e., showed a small efflux of intracellular Cl^−^ (Figure 7A). On average, the mean amplitude of “high K^+^”-induced pH_i_ responses was 0.33 ± 0.06 (*p* < 0.05, *n* = 5), i.e., about 2 times higher than in the control ACSF, while the mean change of [Cl^−^]_i_ was −0.14 ± 0.03 (*p* < 0.05, *n* = 5), i.e., about 30 times smaller in comparison with control conditions. After washing with normal ACSF, the base levels of pH_i_ and [Cl^−^]_i_ returned to initial values; in addition, the amplitudes of K^+^-induced transients were fully recovered (Figure 7B,C).

These results are in accord with the above-described effect of lowering external Cl^−^ on synaptically induced changes of pH_i_ and [Cl^−^]_i_.

### 2.9. Changes in [Cl^−^]_i_ and [H^+^]_i_ during [K^+^]_o_-Induced Depolarization in the Low-Na^+^ ACSF

Extracellular Na^+^ is one of the important participants and regulators of transporters of intracellular proton concentration [1]. For instance, the electrogenic sodium bicarbonate cotransporter NBCe1 (SLC4A4) contributes to intracellular as well as extracellular acid/base homeostasis in the brain and its dysfunctions are associated with pathophysiological states [16,41,42]. In addition, the Na^+^-dependent 2HCO_3_^−^ /Cl^−^-exchanger (NDCBE) represents an important pH- Cl^−^-coupled physiological controller of ions [15].

Lastly, we analyzed how a decrease in external Na^+^ affects changes in pH_i_ and [Cl^−^]_i_ caused by high K^+^ depolarization. Similar to the above-presented results, in ASCF containing normal Na^+^ concentration, K^+^-induced depolarization caused intracellular elevation of both H^+^ and Cl^−^ ions (Figure 8A). On average, the increase in [Cl^−^]_i_ was 12.06 ± 1.19 mM (Figure 8B) and acidification was 0.18 ± 0.04 pH (Figure 8C).

Changing the solution to a low-Na^+^ ASCF (replacement of NaCl with choline Cl) led to a decrease in the base level of intracellular Cl^−^ and H^+^ (Figure 8A). The mean changes were for [Cl^−^]_i_ by 1.98 ± 0.15 mM (*n* = 5, *p* < 0.05, 6 months) and for H+ by 0.08 ± 0.01 pH units (*n* = 5, *p* < 0.05, 6 months). In low-Na^+^ ASCF, responses to K^+^-induced depolarization were potentiated by more than 2 times for both ions, resulting in the mean changes of 25.77 ± 2.74 mM and 0.40 ± 0.05 pH units (*n* = 5, Figure 8B,C).

## 3. Discussion

Our study provides evidence that the optosensoric reporter, ClopHensor, expressed in transgenic mice, represents an efficient tool for non-invasive monitoring of intracellular Cl^−^ and H^+^ ions. Applying different conditions for modulation of neuronal activity in brain slices from transgenic mice, we demonstrated the ability of ClopHensor to simultaneously monitor and analyze changes in [Cl^−^]_i_ and [H^+^]_i_ during artificial changes of its equilibrium.

Maintaining physiologically relevant concentrations of H^+^ and Cl^−^ has a pivotal role in controlling neuronal excitability in the adult brain and during development, and is likely to be crucial in pathophysiological conditions. Both ions play an important role in many cellular processes, including neurotransmission, regulation of membrane potential, cell volume and water–salt balance, modulation of the functions of different proteins, including voltage-gated and receptor-operated channels [1,12,43,44,45,46]. Because of high metabolic activity, accompanied by depolarization, neurons may be susceptible to acidification-induced injury, as can happen in excessive synaptic activation or during conditions of anoxia or ischemia.

The steady-state physiologically relevant dynamic range of pH_i_ and [Cl^−^]_i_ is determined by the balance of both passive transport, through ion channels and by active mechanisms via exchangers or cotransporters [1,46,47].

The intracellular Cl^−^ in neurons of the mammalian brain is primary regulated by two cation–Cl^−^ cotransporters, the Na^+^/K^+^/2Cl^−^ cotransporter 1 (NKCC-1) and the K^+^/Cl^−^ co-transporter 2 (KCC-2) [46,47,48,49]. NKCC1 pumps Cl^−^ into neurons, while KCC2 uses the energy of the K^+^ gradient to extrude Cl^−^ from neurons and maintain relatively low [Cl^−^]_i_ (Figure 9). Dysfunction or changes in the expression of these cotransporters is critically linked to the etiology of several neurologic disorders including epilepsy, acute trauma, ischemia, autoimmune disorders, and neuropathic pain [48,50,51].

Similarly, a modest shift of intra- and extracellular H^+^ ion concentration can have significant effects on brain functions, such as neuronal excitability, synaptic transmission and metabolism [1,52]. This is the result of the sensitivity of a large number of processes to protons, such as ion channel gating and conductance, synaptic transmission, cell-to-cell communication via gap junctions, and enzymatic activity in brain energy metabolism [52,53,54,55]. The major pH regulating transporters identified in the mammalian brain so far comprise the Na^+^/H^+^ exchanger (NHE1), electrogenic Na^+^/HCO_3_^−^ cotransporter 1 (NBCe1), Na^+^-dependent Cl^−^/HCO_3_^−^ exchange (NDCBE), and Na^+^-independent anion exchanger (AE3) (Figure 9; [1,15,56]. NBce-1, NDCBE and AE3 transporters perform translocation of HCO_3_^−^, thus controlling the physiological range of pH_i_. The electrogenic Na^+^/HCO_3_^−^ cotransporter NBCe1 is one of the major regulators of [H^+^]_i_ and is expressed in most brain cell types, with the most prominent expression being astrocytes [1]. It has been shown that NBCe1 is not only an acid extruder/base loader, as suggested in early studies [57,58], but also an acid loader/base extruder [16]. Anion Cl^−^/HCO_3_^−^ exchange (AE3) is known as an important acid-loading performer in brain cells [16,59]. The finding that hippocampal neurons of knockdown AE3 mice (Ae3^–/–^) lack detectable Cl^−^/HCO_3_^−^ exchange activity [60] suggests that AE3 plays a critical role in the maintenance of the Cl^−^ equilibrium potential and/or pH_i_ in neurons.

The role of monocarboxylate transporters (MCTs) in the regulation of the functional integrity of synaptic transmission has been demonstrated. In excitatory synapses, MCT constitutively supports synaptic transmission, even under conditions when there is a sufficient amount of glucose and intracellular ATP [61]. Monocarboxylates cause a decrease in the pH of neurons, which is associated with changes in bioelectric activity [62]. MCT’s involvement in Cl modulation requires future analysis.

These features of transporters emphasize the need for simultaneous monitoring of [Cl^−^]_i_ and [H^+^]_i_ when analyzing the mechanisms of ion homeostasis and neuronal activity, particularly using neuropathological models. A genetically encoded sensor, named ClopHensor, was proposed for simultaneous measurement of Cl^−^ and H^+^ ion concentrations [27]. This construct has shown its effectiveness at the heterologous expression in cell lines, demonstrating good sensitivity and high stability to bleaching during long fluorescence measurements [28]. We recently presented a line of transgenic mice expressing ClopHensor in neurons and obtained a detailed map of its distribution in the mouse brain [32]. Expression of this probe in transgenic mice was found to be highly specific and reproducible in different animals, suggesting that this experimental model represents a promising tool for analysis of dynamic changes in [Cl^−^]_i_ and pH_i_.

In the present study, we analyzed the effectiveness of ClopHensor expressed in transgenic mice for monitoring [Cl^−^]_i_ and [H^+^]_i_ in neurons of hippocampal slices upon changing the ionic equilibrium by pharmacological modulation of neuronal activity, changes in extracellular concentrations of Ca^2+^, Cl^−^ or Na^+^, and by depolarization caused by high-frequency synaptic stimulation or the use of high K^+^.

### 3.1. Kinetics of Synaptically Induced Transients of [Cl^−^]_i_ and [H^+^]_i_

High frequency induced stimulation of Schaffer’s collaterals resulted in the elevation of [Cl^−^]_i_ and [H^+^]_i_ and slow recovery. Importantly, the decay kinetics of Cl^−^ transients was nearly 5-fold faster than pH. The experiments on wild type mice using the chemical pH sensor, BCECF, confirmed remarkably slow kinetics of pH_i_ recovery. These differences in the kinetics of recovery of Cl^−^ and H^+^ responses may reflect the involvement of different cellular participants in control of these ions’ homeostasis.

In the study using the ClopHensorN indicator heterologously expressed in cultured hippocampal slices, about 1.5 times faster decay of intracellular Cl^−^ concentrations were indicated [29]. A profound analysis is necessary to reveal relationships and efficacy of different transporters and other mechanisms involved in the recovery of [Cl^−^]_i_ and [H^+^]_i_ from synaptically induced disequilibrium.

### 3.2. Effect of GABA Receptor Inhibition on Synaptically Induced Transients of [Cl^−^]_i_ and [H^+^]_i_

Results of our study demonstrate that in hippocampal slices of transgenic mice, high frequency stimulation of Shaffer’s collaterals causes an intracellular increase of both ions, Cl^−^ and H^+^. Synaptic stimulation causes the release of neurotransmitters, primary glutamate and GABA, which activates corresponding receptors, leading to the opening of cation and anion-selective ion channels and resulting depolarization of neurons.

As predicted, inhibition of GABA-mediated chloride conductance by bicuculline resulted in a decrease of the synaptically induced influx of Cl^−^. On the other side, this decrease was accompanied by potentiation of acidific [H^+^]_i_ transients. This could be a consequent of two main things: (i) increased synaptic stimulation due to blockade of inhibitory GABA receptors; (ii) changes in the activity of transporters.

In support of the first possibility, the electrophysiological analysis showed that bicuculline causes an elevation in the amplitude of evoked local field potentials. On the other side, experiments that lowered the extracellular Cl showed that in spite of the strong reduction of evoked local field potentials, the amplitude of synaptically induced pH responses was potentiated nearly 10 times in comparison with control.

We suggest that this may be a consequence of the weakening of the acidifying activity of the Cl^−^HCO_3_^−^ exchange. This assumption was tested in experiments with high K^+^-induced depolarization and changes of external concentrations of ion participating in the functioning of transporters.

### 3.3. Analysis of High K^+^-Induced Depolarization on Changes of [Cl^−^]_i_ and [H^+^]_i_ in External Ca^2+^-Free Conditions

Depolarization induced by application of high K^+^ or by other means has been used in a number of studies for analysis of pH_i_ changes in cultured and freshly dissociated neurons, and brain slices [37,38,39,40,63]. In most studies, depolarization induced by application of high external K^+^ was accompanied by elevation of intracellular Ca^2+^ and decrease of pH, i.e., acidification [37,39]. Early observations performed on cultured cells showed that under external Ca^2+^-free conditions, the acidification responses were either completely prevented [38] or even inversed, i.e., exhibited an increase in pH_i_ [39]. Moreover, a blocker of voltage-gated Ca^2+^ channels inhibited K^+^-induced responses, suggesting that an increase in [Ca^2+^]_i_ is a key factor for acidosis to occur [38].

We tested this suggestion in experiments on brain slices of ClopHensor-expressing mice and obtained distinct results. In our experiments, transition to external Ca^2+^-free conditions alone caused a decrease in base pH, i.e., acidification. Surprisingly, K^+^-induced depolarization produced additional acidification. The responses were only partially attenuated in comparison with control, suggesting an only partial contribution of Ca^2+^-dependent processes in pH_i_ responses induced by membrane depolarization.

The reasons for these contrast observations have to be clarified. Among them may be the use of HEPES solution in cell culture experiments and different energy states of neurons.

### 3.4. Effect of Decreasing Extracellular Cl^−^ and Na^+^ on K^+^-Induced Changes in [Cl^−^]_i_ and [H^+^]_i_

#### 3.4.1. Decreasing Extracellular Cl^−^

In our experiments, decreasing Cl^−^ in ASCF resulted in a slow-developing (10–15 min) strong decrease in the base level of cytoplasmic Cl^−^ and elevation of pH, i.e., extruding of H+ from neurons. However, in “low-Cl^−^” conditions, high K^+^-induced depolarization caused remarkable potentiation of acidification pH responses, while Cl^−^ responses were nearly completely suppressed. These observations are in accordance with results obtained at high-frequency synaptic stimulation and indicate that different mechanisms are involved in the control of [Cl^−^]_i_ and pH_i_ at changes of extracellular ion concentrations and during depolarization of neurons. A decrease in extracellular Cl^−^ weakens the ability of the Cl^−^/HCO_3_^−^ transporter to pump Cl^−^ into neurons in exchange for HCO_3_^−^, which leads to slow alkalinization. At high K^+^ application, depolarization induces elevation of intracellular Ca^2+^, resulting in stimulation of PMCA transporter activity and consequent acidification.

#### 3.4.2. Decreasing Extracellular Na^+^

Since the NHE, NBCe1 and NDCBE transporters operate on a Na^+^ gradient, a decrease in the extracellular Na^+^ concentration should lead to a decline in the operation of these pumps and, as a result, to a weakening of protons extruding from the cytoplasm, and a decrease in intracellular HCO_3_^−^ We assume that for these reasons, in our experiments, there was a decrease in the pH_i_ of neurons during the transition to low Na^+^ and a potentiation of acidic responses to depolarization caused by increased K^+^. In the future, these processes will be analyzed in detail.

In conclusion, the baseline level and fluctuations of intracellular Cl^−^ and pH play a crucial role for intercellular and intracellular signaling, as well as for cellular and synaptic plasticity. Our study demonstrates that transgenic mice expressing ClopHensor provide ample opportunities for studying the homeostasis of chloride and hydrogen not only under normal conditions, but also in pathology.

In particular, Cl^−^ gradients are disrupted in epilepsy, especially in early childhood, which significantly complicates treatment. Hydrogen gradients in the central nervous system are disrupted by neuroinflammatory processes of various origins and ischemia. Research in these areas is relevant today, and ClopHensor mice can be a reliable tool for in-depth analysis of the mechanisms controlling the physiological ranges of concentrations of these ions.

## 4. Materials and Methods

### 4.1. Animals

Experiments were conducted on laboratory ICR CD-1 outbreed mice of both sexes at postnatal days 10–12 and adult transgenic mice, aged 2–6 months, strain C57BL/6N expressing ClopHensor (Diuba et al., 2020). Use of animals was carried out in accordance with the Guide for the Care and Use of Laboratory Animals (NIH Publication No. 85–23, revised 1996) and European Convention for the Protection of Vertebrate Animals used for Experimental and other Scientific Purposes (Council of Europe No. 123; 1985). All animal protocols and experimental procedures were approved by the Local Ethics Committee of Kazan State Medical University. Mice had free access to food and water and were kept under natural day length fluctuations. Animals were not involved in any previous procedures.

### 4.2. Solutions and Drugs

Brain slices were prepared in ice-cold high potassium solution containing (in mM): K-gluconate 120, HEPES acid 10, Na-gluconate 15, EGTA 0.2, NaCl 4 (pH 7.2, 290–300 mOsm). After cutting, slices were incubated in a high magnesium artificial cerebrospinal fluid (ACSF) containing (in mM): NaCl 125, KCl 2.5, CaCl_2_ 0.8, MgCl_2_ 8, NaHPO_4_ 1.25, glucose 14, NaHCO_3_ 24 (pH 7.3–7.4, 290–300 mOsm). Storage of slices and performing of experiments were conducted in ACSF, containing (in mM): NaCl 125, KCl 2.5, CaCl_2_ 2.3, MgCl_2_, 1.3, NaHPO_4_ 1.25, glucose 14, NaHCO_3_ 24 (pH 7.3–7.4, 290–300 mOsm). The ACSF was continuously oxygenated with 95% O_2_ and 5% CO_2_ to maintain the physiological pH. The following drugs were used: APV (40 μM, Hello Bio, Bristol, UK, Cat# HB0225), CNQX (10–40 μM, Sigma Aldrich, St. Quentin Fallavier, France, CAS: 115066-14-3), (-)-Bicuculline methochloride (10 μM or 40 μM, Tocris, Science Park Abingdon, UK, Cat#0131), BCECF/AM (10 μM, Sigma Aldrich, St. Quentin Fallavier, France, CAS: 117464-70-7). All drugs were diluted on the day of the experiment from the 1000× to 4000× stocks kept at −20 °C.

### 4.3. Preparation of Brain Slices

Sagittal 350 μm thick sections of the cerebral hemispheres containing the hippocampus were obtained with the use of a vibratome of Model NVSLM1, World Precision Instruments. Mice were euthanized by decapitation. The brain was quickly removed and placed in a Petri dish filled with ice-cold high-K^+^ solution. The cerebellum and olfactory bulbs were cut off using a scalpel; the cerebral hemispheres were separated by cutting along the longitudinal fissure and mounted onto the vibratome specimen disc using superglue, orienting them downward with the sagittal cut surface and the cortex facing the razor blade. Sagittal 350 μm thick sections were prepared in ice-cold high K^+^ solution. After being cut, slices were incubated for 15 min at room temperature in a resting high magnesium oxygenated ACSF. Then, slices were placed in a chamber filled with oxygenated ACSF. Before use, slices were allowed to recover for at least 1 h at room temperature.

Experiments were conducted during the period of 1–6 h after slicing. For the recordings, the brain slices were placed in a chamber perfused with an oxygenated ACSF. Recordings were carried out at 30–31 °C with the speed of perfusion of 25 mL/min.

### 4.4. Fluorescence Imaging in Brain Slices

#### 4.4.1. Monitoring of [Cl^−^]_i_ and pH on Brain Slices ClopHensor Mice

Fluorescence images were obtained using an upright microscope Olympus BX51WI equipped with the iXon Life 897 EMCCD camera (Andor, Oxford Instruments, Abingdon, UK), a 4-Wavelength LED Source (LED4D001, Thorlabs, Newton, NJ, USA) accompanied with a Four-Channel LED Driver (DC4100, Thorlabs), a quad-band filter set (Cat# 9403, Chroma, Foothill Ranch, CA, USA), and a water-immersion objective (60× magnification, 1 numerical aperture; LumPlanFL N, Olympus, Tokyo, Japan) (Figure 10).

The 4-Wavelength LED Source was supplied by light-emitting diodes (LEDs) at 365 nm, 455 nm, 505 nm and 590 nm, the last three of which were used in our study. The capacity of lighting and the LED switching order were adjusted via the DC4100 driver. Cyan channel fluorescence (455 nm excitation) was detected from 460 nm to 485 nm, green channel (505 nm excitation) from 527 nm to 551 nm, and red channel (590 nm excitation) from 600 nm to 680 nm.

All peripheral hardware control, image acquisition and the average fluorescence intensity measurement were achieved using DriverLed software (KSMU, Kazan). Regions of interest (ROI) were set around pyramidal cell bodies of the CA1 hippocampal region. The average fluorescence intensity of the ROI was tracked online at the time of the real-time fluorescence imaging. Numerical data were output by the DriverLed software on the computer in the form of text documents, which were subsequently parsed using Excel 2016 (Microsoft).

Excitation of E2GFP was provided by light-emitting diodes (LEDs) at 455 nm and 505 nm, and a LED at 590 nm provided excitation of DsRed. The duration of excitation at each wavelength was usually 70–110 ms. Examples of the images taken are shown in Figure 1C. Fluorescent emission was recorded continuously from the hippocampal CA1 pyramidal cells with a sampling interval of 10 s. Evoked emission signals were obtained by stimulation of Schaffer’s collaterals by glass bipolar electrode, filled with ACSF, placed in the stratum radiatum of the CA2 hippocampal area (100 Hz for 20 s, 20–320 μA, single pulse width 200 μs).

#### 4.4.2. Monitoring of pH Using BCECF

As an additional control of proper pH monitoring of intracellular pH in hippocampal neurons, we used the pH-sensitive dye, 2′,7′-Bis (2-carboxyethyl)- 5 (6)-carboxyfluorescein-acetoxymethyl ester (BCECF-AM) (Figure 11A). The fluorescent dye, BCECF, was introduced for measuring cytoplasmic pH by Roger Tsien and co-workers [36]. Presently used BCECF-AM is a mixture of three types of cell-permeable non-fluorescent molecules, which are converted to fluorescent non-membrane-penetrating form by intracellular esterases [21]. This dye provides dual-excitation ratiometric monitoring of intracellular pH.

We determined the pH changes by calculating the ratio of two emission signals obtained after illumination by light at 455 nm and 505 nm (Figure 11B). The intracellular pH changes after tetanic stimulation and its modulation by bicuculline were analyzed. Experiments were conducted on wild type juvenile mice (P10-P12). Registration conditions were similar to those performed on slices from transgenic ClopHensor mice. Evoked pH-specific ratiometric emission signals were obtained by giving the same pattern of electrical stimulation (100 Hz for 20s); the stimulus intensities ranged from 200 μA to 250 μA.

Hippocampal slices were stained with 10 µM BCECF-AM for 40 min in the oxygenated ACSF. Then the dye was washed out and slices were superfused with normal ACSF for at least 40 min to allow esterases to cleave AM and stabilise pHi. For fluorescent monitoring, slices were transferred to the optical recording chamber, which was mounted on the stage of an upright microscope (Olympus BX51WI) (Figure 10).

The setup consisted of a microscope (Olympus BX51WI equipped with the iXon Life 897 EMCCD camera (Andor camera)), a 4-Wavelength Thorlabs LED Source (Light source), and a quad-band filter set (excitation filter). Excitation light with wavelengths of 505 nm, 455 nm and 590 nm, each for 70–100 ms, was sequentially applied to slices and the emission signals from CA1 pyramidal cells were registered by an EMCCD camera. The stimulation electrode was placed in the stratum radiatum of the CA2 hippocampal area. The evoked local field potentials were recorded using a glass micropipette electrode, placed in the stratum radiatum of the CA1 hippocampal area and an HEKA EPC 10 Patch Clamp Amplifier (not shown).

Slices were alternately illuminated at 455 nm and 505 nm with duration of 100 ms. Light from both wavelengths was passed through appropriate excitation and emission filters. Fluorescence image pairs were captured every 10 s by an intensified camera. Evoked emission signals were obtained in the same way as for ClopHensor: by sustained high-frequency electrical stimulation of Schaffer’s collaterals (100 Hz for 20 s, 20–320 μA, single pulse width 200 μs).

### 4.5. Electrophysiological Recording

Evoked local field potentials (eLFP) were recorded in the stratum radiatum of the CA1 hippocampal region using glass micropipette electrodes filled with ACSF (resistance 1–2 MΩ) and an HEKA EPC 10 Patch Clamp Amplifier (HEKA Elektronik, Lambrecht, Germany). The DS3 Constant Current Isolated Stimulator (Digitimer, Welwyn Garden, UK) and a bipolar stimulating electrode placed on the Schaffer’s collaterals at the hippocampal area CA2 were used for the induction of eLFPs. Current pulses 20–300 μA in amplitude and 200 μs in duration were applied to obtain reliable eLFPs. Single eLFPs were recorded continuously every 20 s. PatchMaster software (HEKA Electronik, Lambrecht, Germany) was used to record eLFPs, control the HEKA EPC 10 Patch Clamp Amplifier and the DS3 Constant Current Isolated Stimulator.

### 4.6. Intracellular pH and Cl^−^ Calibration of ClopHensor on Hippocampal Slices

To perform pH and Cl^−^ calibration, we used a double ionophore technique [64]. The brain slices were exposed to the antibiotic nigericin, which acts as H^+^/K^+^ antiporter and the Cl^−^/OH^−^ antiporter tributyltin, which forms pores in the cell membrane and allows external Cl^−^ to equilibrate with intracellular Cl^−^ [65]. For the action of the compounds, before fluorescent monitoring, the hippocampal slices were kept for 1–28 h at +4 °C in the following solution: 150 mM K-Gluconate, 20 mM HEPES, and 10 mM D-glucose) with addition of nigericin (20 mkM, pH = 7.22–7.25) and tributyltin (20 mkM).

For pH calibration, the solutions with different pH values (from 6.23 pH to 8.04 pH) were prepared by adding KOH for alkalization. For calibration of Cl^−^, the solutions were prepared at pH = 7.25 and contained a different concentration of Cl^−^ (0, 3, 10, 30, 100 mM). To keep the osmolality of the solutions, KCl was correspondently substituted by the K-gluconate (150, 147, 140, 120, 50 mM). All of these solutions contained 20 mM HEPES, and 10 mM D-glucose.

Fluorescent signals were recorded from the CA1 zone of the hippocampus with the following lighting parameters in the Drive LED program: Exposition 70 ms, LED on 100 ms, binning 2, Voltage (590 nm) 70 mV, Voltage (505 nm) 10 mV, Voltage (455 nm) 15 mV. The slices were incubated in each solution until the fluorescence stabilized, usually for 20–30 min.

We obtained a linear dependence for pH (from 6.7 pH to 7.6 pH) on the fluorescence ratio (F_505nm_/F_455nm_):pH = F_505nm_/F_455nm_ × K_1_ + K_2_,(3)
where K_1_ = 0.44, K_2_ = 6.06.

Coefficients were obtained by fitting data for 6 slices.

To estimate the [Cl^−^]_i_ from calibration data obtained on 4 slices, the following equation was used:(4)$${\left[Cl^{-}\right]}_{i}=K_{0}+\frac{K_{1}-K_{0}}{1+{\left(\frac{K_{3}}{F_{590\;\mathrm{nm}}/F_{455\;\mathrm{nm}}}\right)}^{K_{2}}}$$
where K_0_ = −0.33, K_1_ = 33.62, K_2_ = 10.1, K_3_ = 0.87.

All coefficients were obtained by fitting data using the Igor Pro 6.02 software.

### 4.7. Data Analysis and Statistics

Amplitudes of evoked local field potentials were measured using the Online Analysis function of PatchMaster software (HEKA Electronik, Lambrecht, Germany). Superimposed average traces of evoked local field potentials were generated in PatchMaster software and processed for further presentation in Igor Pro 6.02 software (WaveMetrics, Tigard, OR, USA).

The average emission intensity of the region of interest (ROI) was measured online at the time of the real-time fluorescence imaging or, if necessary, recalculated using DriverLed software. Excel 2016 (Microsoft) software was used to compute and plot Cl^−^ and pH-dependent ratios of emission intensities and to measure the amplitudes evoked by tetanic stimulation pH- and Cl^−^-specific ratiometric fluorescence emission signals.

Origin 15 software was used to perform a statistical analysis of the data, to plot the graphs and compute the decay times of the evoked pH- and Cl^−^ -specific fluorescence emission signals.

Data are represented as means ± SEM. Statistical significance was determined using Paired Sample Wilcoxon Signed Rank Test and Mann Whitney U test. Differences were considered significant at *p* < 0.05.

## Figures and Tables

**Figure 1 ijms-22-13601-f001:**
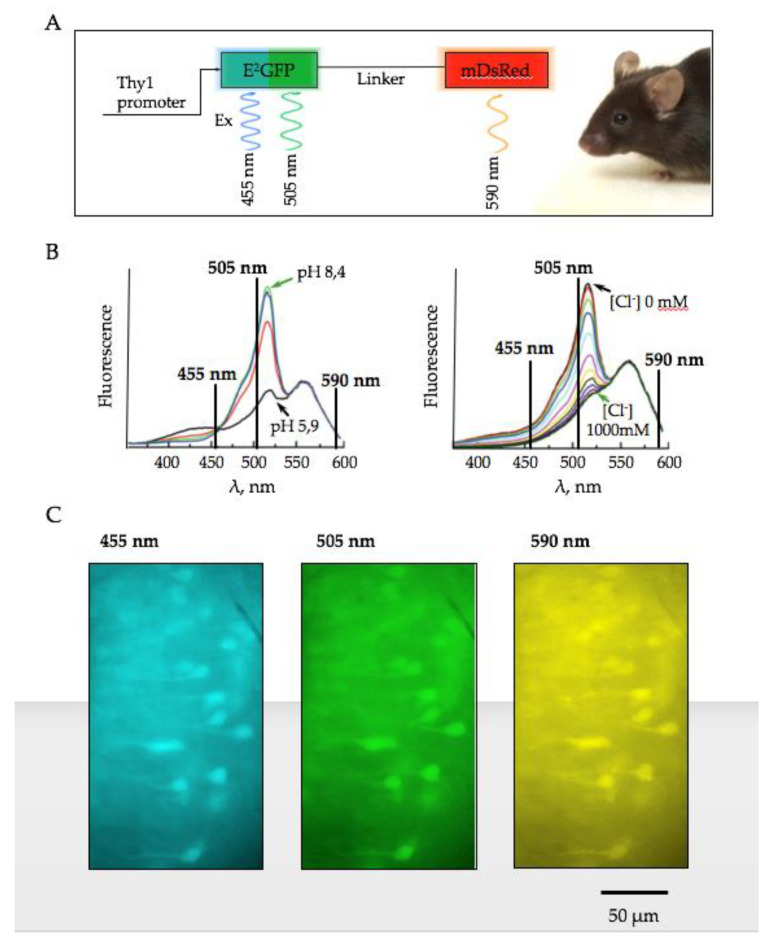
Monitoring pH and Cl^−^ in brain slices of mice expressing ClopHensor. (**A**) Schematic representation of ClopHensor. (**B**) Excitation spectra of ClopHensor collected at different pH values (5.9, 6.9, 7.4, and 8.4) in the absence of Cl^−^ (left) and at increasing Cl^−^ concentration (0–1 M) and constant pH = 6.9 (right) (modified from Arosio et al., 2010 [27]). (**C**) Micrographs of pyramidal cells of CA1 hippocampal region, expressing ClopHensor, under illumination with excitation light with wavelengths of 455 nm, 505 nm and 590 nm (age P7).

**Figure 2 ijms-22-13601-f002:**
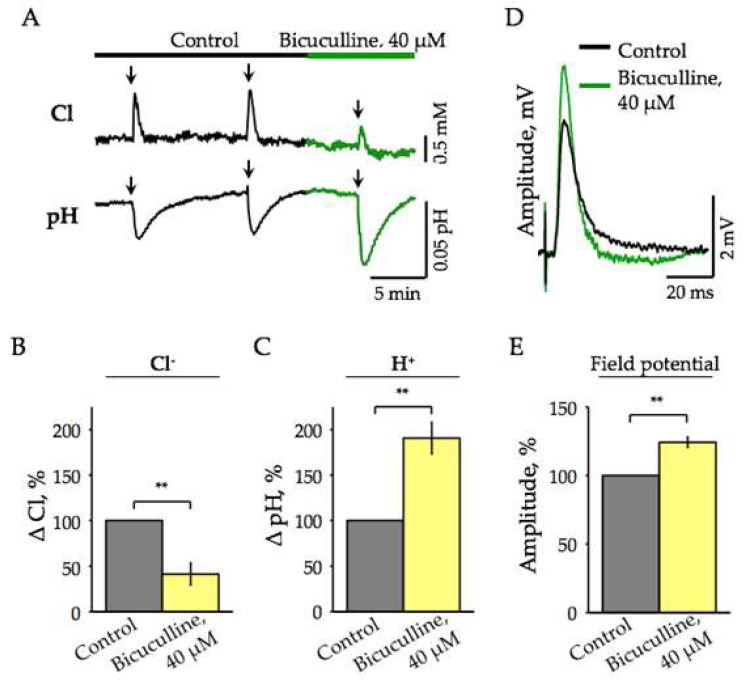
Effect of bicuculline on Cl^−^- and pH-transients induced by synaptic stimulation of hippocampal CA1 neurons in brain slices of transgenic ClopHensor mice. (**A**) Traces of Cl^−^- (top) and pH-specific emission signals (bottom) in control (black) and the presence of 40 μM bicuculline (green). Arrows indicate stimulation (100 μA, 100 Hz, 20 s). (**B**) Mean inhibition by bicuculline (40 μM) of Cl^−^ transients induced by synaptic stimulation of CA1 hippocampal neurons (mean percentage ± SEM, *n* = 7, 2–8 months). ** Significant difference with *p* < 0.01 (Paired Sample Wilcoxon Signed Rank Test). (**C**) Mean potentiation by bicuculline (40 μM) of stimulation-induced pH-transients. Summary data from 8 slices (mean percentage ± SEM, *n* = 8, 2–8 months). ** Significant difference with *p* < 0.01 (Paired Sample Wilcoxon Signed Rank Test). Age of mice 2–8 months. (**D**) Superimposed traces of evoked local field potentials (eLFP) induced by single stimulation of Schaffer’s collaterals in control (black) and the presence of 40 µM bicuculline (green). Presented averaged traces of 10 individual eLFPs (stimulation: 20 μA, single pulse width 200 μs; age 7 months). (**E**) Summary of the eLFP amplitude potentiation by 40 µM bicuculline. 100% is the amplitude of eLFPs in control condition. Values are mean ± SEM (*n* = 7). ** Significant difference with *p* < 0.01 (Paired Sample Wilcoxon Signed Rank Test). Age of mice 2–8 months.

**Figure 3 ijms-22-13601-f003:**
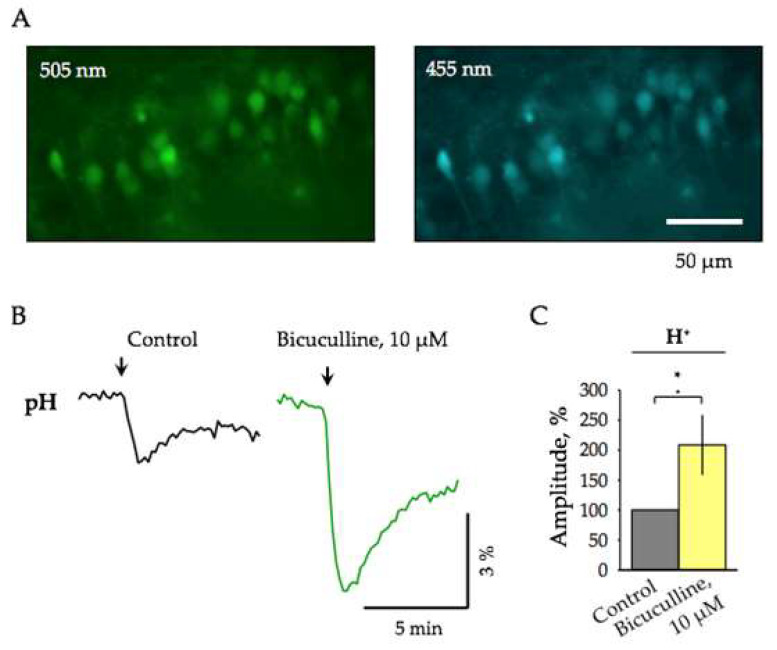
The effect of bicuculline on the evoked pH signals recorded using BCECF-AM. (**A**) Micro-graphs of pyramidal cells of the CA1 hippocampal region, loaded by BCECF-AM, under illumination with excitation wavelengths of 505 nm and 455 nm (P12). (**B**) Representative traces of pH-specific ratiometric emission transients evoked by high-frequency electrical stimulation (200 μA, 100 Hz for 20s) in control (left trace, black) and after addition of 10 µM bicuculline (right trace, green). The arrows indicate the moments of stimulation (P11). (**C**) Summary of bicuculline action on the amplitude of pH changes induced by synaptic stimulation. Normalization to the control values (mean ± SEM, P10–12, *n =* 5). * Significant difference with *p* < 0.05 (Paired Sample Wilcoxon Signed Rank Test).

**Figure 4 ijms-22-13601-f004:**
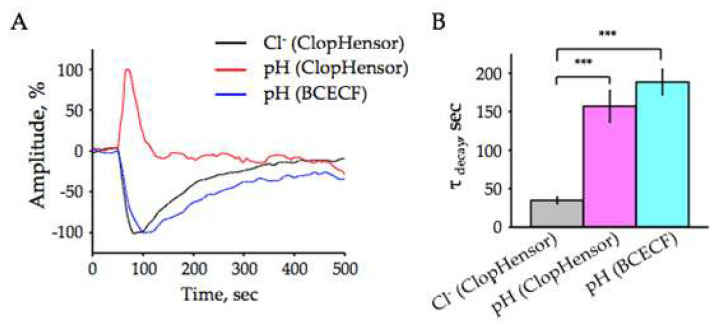
Comparison of the decay kinetics of stimulation-induced Cl^−^ and pH transients in hippocampal CA1 neurons. (**A**) Representative traces of normalized Cl^−^-(red trace) and pH-(black trace) transients registered using the transgenic ClopHensor mouse (stimulation 100 Hz for 20 s; age 6 months), and pH-specific signals using BCECF-AM (blue trace; stimulation: 100 Hz for 20 s; age P11). (**B**) The mean time constants of decay of the Cl^−^ and pH stimulation-induced responses were recorded with ClopHensor (age 2–7 months) and BCECF (age P10-12). Data from 9–16 slices. Values are mean ± SEM. *** Significant difference with *p* < 0.001 (Mann Whitney U test).

**Figure 5 ijms-22-13601-f005:**
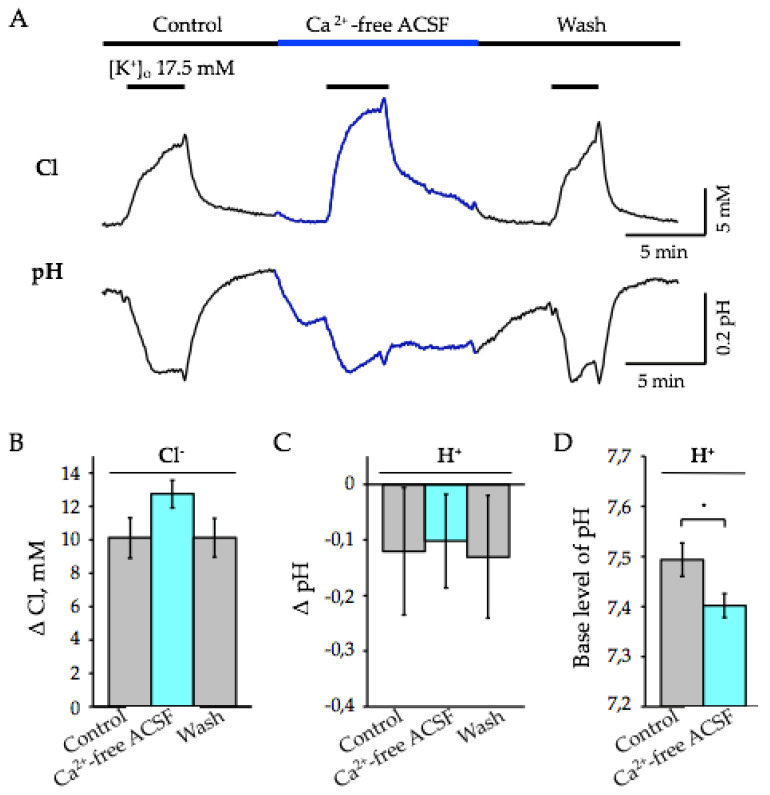
Effect of external Ca^2+^ removal on depolarization-induced changes in [Cl^−^]_i_ and pH_i_ in neurons of hippocampal slices. (A) Typical changes in [Cl^−^]_i_ (top) and pH_i_ (bottom) evoked by the application of 17.5 mM K^+^ in the presence of 2.3 mM external Ca^2+^ (Control) and in Ca^2+^-free ASCF. The periods of K^+^ application and changes in external Ca^2+^ are indicated by the bars above the traces. Note the decrease of pH upon elimination of external Ca^2+^. (**B,C**) The mean [K^+^]_o_-induced changes of [Cl^−^_i_ (**B**) and pH_i_ (**C**) in control and in Ca^2+^-free ASCF. Summary of data from 4 slices (mean percentage ± SEM, age 4 months). (**D**) Mean values of base level of pH_i_ in the control condition (2.3 mM of external Ca^2+^) and in Ca^2+^-free ASCF. * Significant difference with *p* < 0.05.

**Figure 6 ijms-22-13601-f006:**
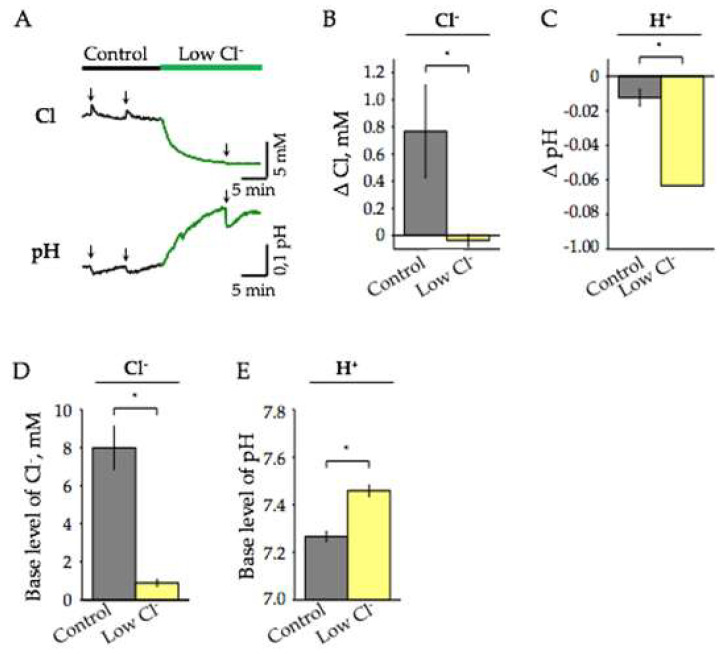
Effect of reduction of extracellular Cl^−^ and high-frequency stimulation on [Cl^−^]_i_ and pH_i_ in the hippocampal CA1 neurons of ClopHensor mice. (**A**) Typical traces of continuous recording of [Cl^−^]_i_ (top) and pH_i_ (bottom) illustrating the effects of the transition to “low Cl^−^” ASCF and responses on high-frequency stimulation (age 6 months). The “low Cl^−^” ACSF application is highlighted by yellow. Arrows indicate moments of stimulation. (**B,C**) Mean changes of synaptically induced [Cl^−^]_i_ (**B**) and pH_i_ (**C**) in conditions of high and low external Cl^−^ (mean ± SEM, *n* = 5, age 6 months). (**D,E**) Summary of the effect of reduction of external Cl^−^ concentration on base levels of [Cl^−^]_i_ (**D**) and pH_i_ (**E**) (mean percentage ± SEM, *n* = 5, 6 months). * Significant difference with *p* < 0.05 (Paired Sample Wilcoxon Signed Rank Test).

**Figure 7 ijms-22-13601-f007:**
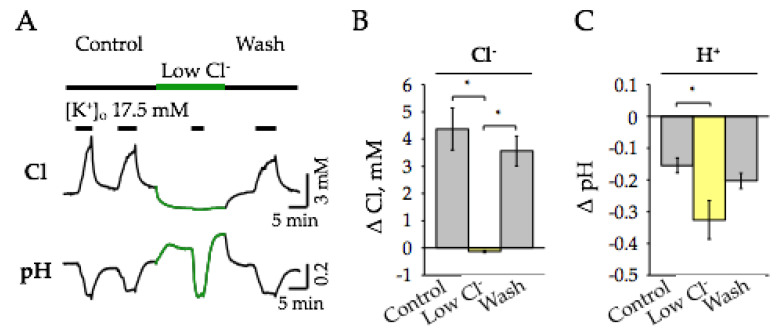
Effect of extracellular Cl^−^ on changes of [Cl^−^]_i_ and pH_i_ caused by K^+^-induced depolarization. (**A**) Typical traces of continuous recording of [Cl^−^]_i_ (top) and pH_i_ (bottom) illustrating the effects of the transition to “low Cl^−^” ASCF and responses to the addition of 17.5 mM K^+^ to hippocampal slices (age 7 months). The “low Cl^−^” ACSF application is highlighted by the green bar. (**B**,**C**) Summary of the effect of K+-induced depolarization on changes of [Cl^−^]_i_ (**B**) and pH_i_ (**C**). Data from 5 slices (mean ± SEM, *n* = 5, 7 months). * Significant difference with *p* < 0.05 (Paired Sample Wilcoxon Signed Rank Test).

**Figure 8 ijms-22-13601-f008:**
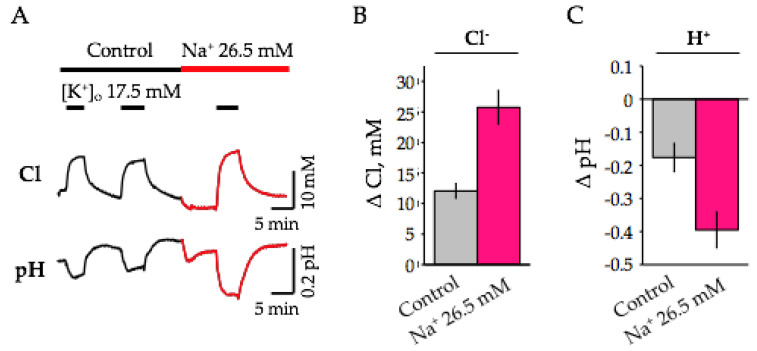
Effect of extracellular Na^+^ on [Cl^−^]_i_ and pH_i_ responses induces by the high K^+^ depolarization. (**A**) Typical traces of continuous recording of [Cl^−^]_i_ (top) and pH_i_ (bottom) illustrating the effects of the transition to “low Na^+^” ASCF and responses to the addition of 17.5 mM K^+^ to hippocampal slices (age 6 months). (**B**,**C**) Summary of the high K^+^-induced changes in concentrations of Cl^−^ (**B**) and H^+^ (**C**) in control (grey columns) and in ACSF containing low Na^+^ (26.5 mM) (red columns). Summary from 5 slices (mean ± SEM, 6 months).

**Figure 9 ijms-22-13601-f009:**
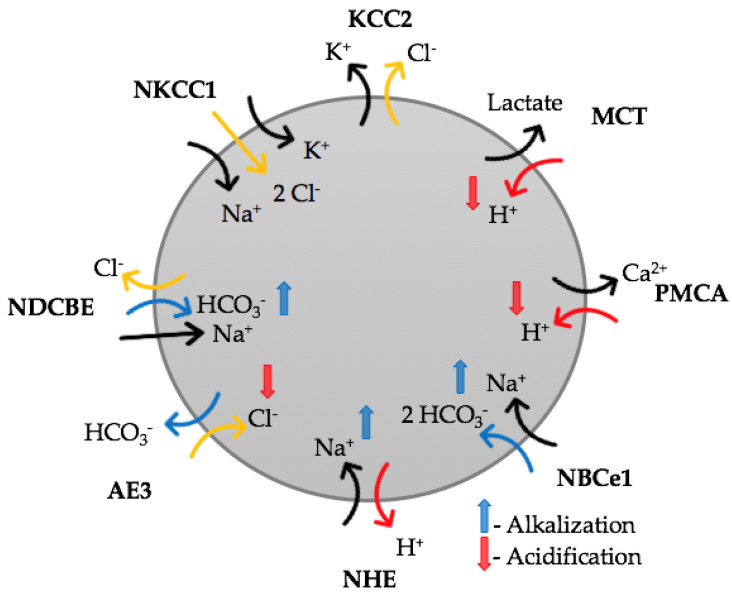
Scheme of key transmembrane transport elements controlling intracellular Cl^−^ and pH in brain cells.

**Figure 10 ijms-22-13601-f010:**
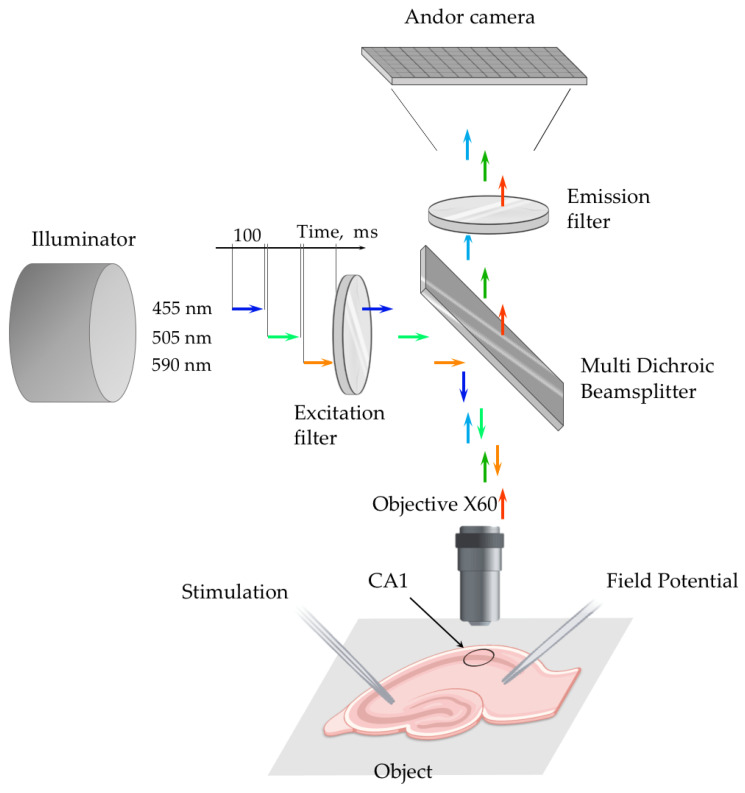
Setup for simultaneous monitoring of pH_i_ and [Cl^−^]_i_ using ClopHensor.

**Figure 11 ijms-22-13601-f011:**
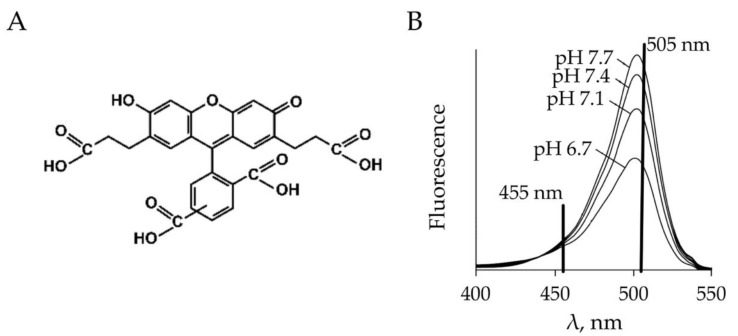
(**A**) The chemical structure of one of the molecules of 2′,7′-Bis (2-carboxyethyl)- 5 (6)-carboxyfluorescein-acetoxymethyl ester (BCECF-AM). (**B**) Excitation spectra of BCECF collected at different pH values. The BCECF pH measurements were made by determining the pH-dependent ratio of emission intensity when it was excited at 505 nm versus the emission intensity when excited at 455 nm.

## Data Availability

Not applicable.

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
