# Peer review of "Simultaneous Monitoring of pH and Chloride (Cl) in Brain Slices of Transgenic Mice"

_ijms, 2021, doi:10.3390/ijms222413601_

Round 1

Reviewer 1 Report

Authors reported a series of experiments in which intracellular chloride and proton concentrations in hippocampal slices were measured simultaneously by using ClopHensor. The work seems to be interesting and maybe promising.  However, to further study and discuss the contributions of each transporters and pumps (PMCA is a plasma membrane Ca2+ pump) as presented in Fig. 9, I hope they should use inhibitors specific for each transporters, such as Furosemide and Bumetanide for NKCC1, and VU0240551 for KCC2.  

There are some grammatical errors observed such as

“As an efficient optosensoric tools in recent years are used genetically encoded proteins…” in the abstract.  

“Among them might be using of HEPES solution in cell culture experiments….(Page 15 line 5)

Authors should check and brush up their manuscript.

There is no explanation about the contribution of MCT in the regulation of intracellular proton concentration in the text.

Author Response

Reply to referee 1.

We are grateful to referee for a critical reading of the manuscript.

We have corrected the text according to your suggestions.

1  ". to further study ...they should use inhibitors specific for each transporter..."

In the future, we plan to use various transporter inhibitors, including DIDS, bumetanide and VU0240551.

  1. "There are some grammatical errors..."

We have made the appropriate corrections and changes in the text (highlighted in red)

In recent years, genetically encoded proteins have been used as effective optosensory means. These probes posess...

Among them may be the use of HEPES solution in cell culture experiments and different energy state of neurons.

  1. "There is no explanation about the contribution of MCT in the regulation
    of intracellular proton concentration in the text."

We added corresponding text in the discussion:

"The role of monocarboxylate transporters (MCTs) in the regulation of the functional integrity of synaptic transmission has been demonstrated. In excitatory synapses, MCT constitutively supports synaptic transmission, even under conditions when there is a sufficient amount of glucose and intracellular ATP ((Ivanov et al., 2011). Monocarboxylates cause a decrease in the pH of neurons, which is associated with changes in bioelectric activity (Bonnet et al., 2018). MCT's involvement in Cl modulation requires future analysis.".

Reviewer 2 Report

In this study, the authors analyze the possibility of non-invasively monitoring the concentration of intracellular ions or the activity of intracellular components using specific biosensors through optosensoristics. To this end, for monitoring intracellular concentrations of chloride ([Cl-] i) and hydrogen ([H +] i) I use a construct, called ClopHensor, which consists of an H +-e Cl---Sensitive variant ofenhanced green fluorescent protein (E2GFP) fused with a monomeric red fluorescent protein (mDsRed), expressed by a line of transgenic mice expressing ClopHensor in neurons.

 The results obtained demonstrate that the transgenic mice expressing ClopHensor represent a reliable instrument for the simultaneous non-invasive monitoring of intracellular Cl- and pH.

The work is very interesting and well organized in its contents.

However, some corrections are needed before a possible publication in this journal.

Materials and Methods section

Pag 15 of manuscript, par4.1. Animals: please specify full name for ICR, in addition, . describe how mice are able to express ClopHensor

4.2. Solutions and drugs and 4.3. Preparation of brain slices: Please, indicate the thickness of Brain slices

The authors state the efficiency of transgenic mice expressing ChlopHensor for a reliable non-invasive monitoring of intracellular Cl- and pH in normal and pathological conditions.

A convincing conclusion is missing from this work, for example what application proposals could there be in the human field?

Author Response

Reply to referee 2.

We are grateful to referee for a critical reading of the manuscript.

The text has been corrected and supplemented in accordance with your suggestions.

  1. P. 15 of manuscript, par4.1. "Animals: please specify full name for ICR, in addition, . describe how mice are able to express ClopHensor ".

We used transgenic mice expressing ClopHensor. The detains of obtaining this line are described in our previous paper (Diuba, et al., 2020). The reference is in the text.

Corrected

  1. "2. 4.2. Solutions and drugs and 4.3. Preparation of brain slices: Please, indicate the thickness of Brain slices"

Corrected

  1. "A convincing conclusion is missing from this work, for example what application proposals could there be in the human field?"

We added the following text in the conclusion:

The baseline level and fluctuations of intracellular Cl- and pH play a crucial role for intercellular and intracellular signaling, as well as for cellular and synaptic plasticity. Our study demonstrates that transgenic mice expressing ClopHensor provide ample opportunities for studying the homeostasis of chloride and hydrogen not only under normal conditions, but also in pathology.

In particular, Cl gradients are disrupted in epilepsy, especially in early childhood, which significantly complicates treatment. Hydrogen gradients in the central nervous system are disrupted by neuroinflammatory processes of various origins and ischemia. Research in these areas is relevant today, and ClopHensor mice can be a reliable tool for in-depth analysis of the mechanisms controlling the physiological ranges of concentrations of these ions.